# Beyond Memorization: Extending Reasoning Depth with Recurrence, Memory and Test-Time Compute Scaling

## Abstract

Reasoning is a core capability of large language models, yet how multi-step reasoning is learned and executed remains unclear. We study this in a controlled cellular-automata (1dCA) framework that excludes memorisation by using disjoint train/test rules. Models are trained on short state sequences, required to *infer* the hidden local rule, and then *chain* it for multiple future steps. We find that most neural architectures learn the rule and achieve high next-step accuracy, but performance drops sharply as the required number of steps increases. Increasing model depth is crucial, and extending *effective* depth via recurrence, memory, or test-time compute improves results but remains bounded. Complementing these controlled experiments, a natural-language proxy game shows that contemporary LLMs largely fail on the complex setting. Together, these results separate genuine rule induction from memorisation, quantify how difficulty scales with reasoning depth, and highlight the joint roles of architecture and training/inference procedures.

## 1 Introduction

Large Language Models (LLMs) demonstrate impressive capabilities in problem-solving and reasoning tasks, e.g., OpenAI's o1 (OpenAI, 2024) and DeepSeek R1 (Guo et al., 2025) models achieved a top-500 ranking in a qualifier for the USA Math Olympiad (AIME). OpenAI system achieved an outstanding result, ranked 6 in the International Olympiad in Informatics (IOI 2025) (OpenAI, 2025). Both Google DeepMind and OpenAI systems achieve gold-medal scores in the International Olympiad in Mathematics (IMO 2025) (Luong & Lockhart, 2025). On the other hand, extensive evidence from ongoing research shows that LLMs still face challenges in multi-step reasoning Dziri et al. (2024); Wan et al. (2024); Holliday & Mandelkern (2024); Gandarela et al. (2024); Mondorf & Plank (2024); Shojaee et al. (2025) and planning Valmeekam et al. (2024), particularly when required to infer and apply underlying rules from data.

These observations raise the following questions:
*1. Is the reasoning exhibited by LLMs the result of genuine generalization, or merely memorization?*
*2. How does task difficulty scale as the required number of reasoning steps increases?*
*3. To what extent do a model's architectural inductive biases, training objectives, and inference procedures limit its reasoning capabilities?*

Transformers (Vaswani et al., 2017) are universal function approximators and, with unbounded depth and precision, are Turing-complete (Cybenko, 1989; Hornik et al., 1989; Dehghani et al., 2019; Yun et al., 2019; Bhattamishra et al., 2020; Pérez et al., 2021; Sanford et al., 2024b). Yet, *finite-depth, fixed-width* models used in practice cannot process arbitrarily long inputs in a single forward pass, and they provably fail on tasks such as graph connectivity, Boolean formula evaluation, and exact arithmetic beyond a bounded length (Merrill et al., 2022; Merrill & Sabharwal, 2023b; Strobl et al., 2023; Feng et al., 2024).

One way to sidestep this depth barrier is to let the model *write its own scratch-pad* of intermediate tokens. Chain-of-Thought (CoT) prompting, process supervision, and reinforcement learning (RL) encourage models to emit multi-step rationales before producing the final answer (Wei et al., 2022; Uesato et al., 2022; Wang et al., 2024; Yao et al., 2024; Kumar et al., 2024). Generating and consuming these extra tokens effectively increases the computational depth in proportion to the

rationale length, enabling transformers to solve dynamic-programming benchmarks (Feng et al., 2024) and to recognize regular languages with linear decoding depth (Merrill & Sabharwal, 2023a). Yet, the main drawback is the need for supervision over intermediate steps, which is expensive or might be unavailable.

A complementary avenue is to *recycle hidden states*. Segment-level recurrence in memory-augmented transformers (Weston et al., 2015; Graves et al., 2014) enables the re-feeding of hidden states across segments (Dai et al., 2019; Rae et al., 2019; Bulatov et al., 2022; Chevalier et al., 2023; Rodkin et al., 2024), whereas state-space models achieve long-range interactions by leveraging linear dynamical systems (Gu et al., 2021; Gu & Dao, 2023). Recurrence deepens the network without emitting extra tokens, but the maximum number of recurrent steps is still limited by the input length. *Adaptive Computation Time* (ACT) (Graves, 2016) removes this upper bound entirely: the model learns to allocate a variable number of layer updates to each token, halting once further computation is predicted to be unhelpful. In principle, ACT grants transformers *unbounded effective depth* while preserving parameter efficiency, which is an appealing property for reasoning tasks that require widely varying amounts of computation.

In this paper, we study *rule abstraction* and *multi-step reasoning* in neural models using a controlled 1D Cellular Automata (1dCA) setting that prevents memorisation by holding out disjoint rule sets between training and test. We cast reasoning as variable-horizon prediction and quantify how architectures and depth-extension strategies cope as the look-ahead $k$ increases. Our main contributions are:

**1DCA-REASONING benchmark.** A variable-length dataset with four task variants (O–S, O–O, O–RS, RO–S) that disentangle rule induction from state propagation; train/test rule sets are disjoint to preclude memorisation.

**LLM evaluation in natural language.** A new *Handsup* task—a worded proxy equivalent to the 1dCA update—used to assess LLMs under varying look-ahead and rule complexity, showing that many LLMs (except Gemini 2.5 Pro) fail on the simplest radius-1 setting.

**Comprehensive architectural comparison.** Side-by-side evaluation of Transformers (GPT-NeoX), LSTMs, state-space models (Mamba), and a memory-augmented Transformer (ARMT) under identical conditions. Fixed-depth (4-layer) models solve $k=1$ but collapse for $k \geq 2$; ARMT extends to $k=2$. We corroborate these trends on a group-multiplication benchmark (Merrill et al., 2024).

**Depth-extension analysis.** With 4-layer backbones: (i) Adaptive Computation Time (ACT) reliably adds $\sim +1$ effective step with modest compute; (ii) GRPO (RL) reaches $k=3$ *without* intermediate supervision; and (iii) token-level Chain-of-Thought attains near-perfect accuracy up to $k=4$.

## 2 METHODS

**Modeling Reasoning with 1d Cellular Automata.** Reason is the capacity of consciously applying logic by drawing valid conclusions from new or existing information.[1] Reasoning about an unfamiliar process naturally splits into two parts: (i) inferring the hidden law that drives state transitions and (ii) chaining that law to predict multiple future steps. One-dimensional cellular automata (1dCA) provide a minimal, fully observable sandbox for this: a local Boolean rule—the toy universe's "micro-physics"—updates each binary state from its neighborhood. In our benchmark the rule is withheld and the train/test rule sets are disjoint, so rote lookup cannot succeed. To solve a task the model must first induce the rule from observed orbits and then apply it repeatedly to roll out future states, cleanly separating genuine rule-based reasoning from mere memorization.

**Background.** An *One-dimensional Cellular Automaton (1dCA)* is a one-dimensional, dynamical system in which space and time are discrete. Let $r \in \mathbb{N} : r \geq 1$ be the *neighborhood radius* in the space represented by a regular lattice of $W \in \mathbb{N} : W \geq 2r + 1$ identical, locally-interconnected *cells* with binary state spaces, $\mathbb{S} = \{0, 1\}$. The 1dCA's *global state*, $x \in \mathbb{S}^W$, is a lattice configuration specified by the values of all states of all cells in the lattice at a given time. This state evolves deterministically in synchronous, discrete time steps according to a *global map* $g_\rho : \mathbb{S}^W \to \mathbb{S}^W$ defined by a *local rule* $\rho : \mathbb{S}^{2r+1} \to \mathbb{S}$, so $[g_\rho(x)]_w = \rho(x_{w-r}, \ldots, x_w, \ldots, x_{w+r})$ (Fig.1a). The sequence of states an 1dCA passes through during its *space–time evolution*, $\mathcal{O}^T(x) = [x, g_\rho(x), g_\rho(g_\rho(x)), \ldots, g_\rho^{oT-1}(x)]$,

---

[1] https://en.wikipedia.org/wiki/Reason

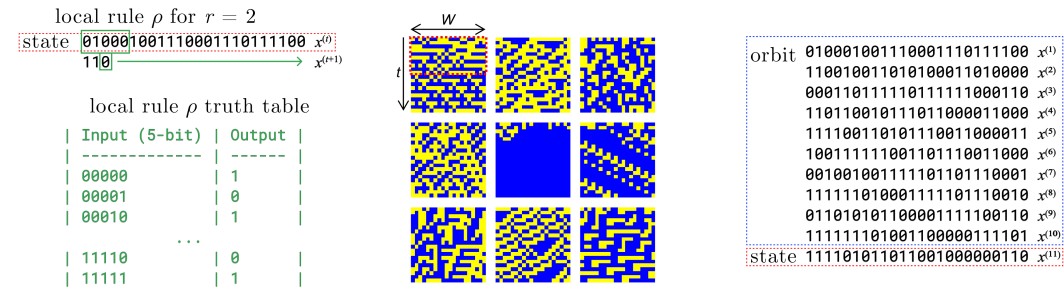

(a) Generation of states in 1dCA    (b) Examples of training samples    (c) Orbit-State (O-S) learning task

Figure 1: **Learning One-dimensional Cellular Automata.** **(a)** Update of state with local rule. **(b)** Orbit of 1dCA is a sequence of binary strings of size $W = 20$. The first $k = 10$ states marked by the red rectangle encode transformer input. **(c)** Given a part of the orbit a model learns to predict the next state (O-S).

defines its *trajectory* or *orbit* from an *initial condition* (configuration) $x$ for $T \in \mathbb{N} : T \geq 1$. Examples of 1dCA orbits are visualized in Figure 1(b).

**Benchmark for reasoning.** Our benchmark instantiates multi-step reasoning with 1dCA trajectories: each example provides a short orbit (e.g., 10 states) generated by a hidden rule; training and test use disjoint rule sets. The model must infer the rule from the observed states and predict future configurations, forcing it to learn a general rule-inference procedure rather than memorize instance-specific mappings. We vary difficulty via *look-ahead* prediction: to give $g_\rho^{oT+k}(x)$ for $k \in \{1, 2, 3, 4\}$ steps ahead (without intermediate states), the model must internally roll out the dynamics, effectively chaining the inferred rule. We call $k$ the *depth of reasoning* and study which architectures can achieve greater depth under this setting.

**Task variants.** The benchmark could emulate the situations when we have supervision on intermediate steps (i.e. the thinking process of the LLM) and when we only have a final look-ahead state. We consider four variations of learning tasks designed to assess different aspects of predictive modeling and rule inference:

*Orbit-State (O-S)*: given an orbit $\mathcal{O}^T(x) = [x^{(1)}, x^{(2)}, \ldots, x^{(T)}]$ where $x^{(1)} \in \mathbb{S}^W$, the objective is to predict the state $x^{(T+k)}$ at look-ahead $k \in \mathbb{N} : k \geq 1$. For $k = 1$ (see Fig.1c) this is a single-step prediction simulating an elementary act of reasoning or a part of a curriculum to learn longer reasoning chain. For $k > 1$ multiple intermediate inference steps are required for the answer.

*Orbit-Orbit (O-O)*: given an orbit $\mathcal{O}^T(x)$ for some $k > 1$ predict the subsequent states up to time $T + k$, generating $\mathcal{O}_{T+1}^{T+k}(x) = [x^{(T+1)}, \ldots, x^{(T+k)}]$. This task simulates step-by-step multi-step reasoning as a learning objective.

*Orbit-State and Rule (O-RS)*: given an orbit $\mathcal{O}^T(x)$ predict the state $x^{(T+k)}$ and the local rule $\rho$. By explicitly optimizing rule prediction, the model receives direct supervision.

*Rule and Orbit-State (RO-S)*: given an orbit $\mathcal{O}^T(x)$ and the local rule $\rho$ predict the state $x^{(T+k)}$ at time $T + k$. Since the rule is explicitly provided, the model can bypass inference of rule structure and focus solely on learning to apply the update.

The rule in our 1dCA setup is based on a neighborhood radius $r = 2$, meaning each bit of the next state depends on a 5-bit window (2 left + current cell + 2 right) from the current state. Since there are $2^5$ possible 5-bit strings, the rule mapping can be represented by a 32-bit string. Each bit in this string corresponds to the output of the rule for a specific input. The position of this output bit within the rule string is determined by the binary value of the 5-bit input (see Fig.1a). For our evaluation we use the exact match metric for state prediction (1 if the state is predicted correctly, 0 if at least one bit is predicted wrong) and bit accuracy (ratio of the correctly predicted bits) for the rule. You can find the examples of training/validation samples in the subsection F.2.

**Neural Models.** In our study, we consider LLMs and small models belonging to several widely-applied architectural families. Long Short-Term Memory (LSTM) networks Hochreiter & Schmidhu-

ber (1997), a class of recurrent neural network (RNN), have proven effective in capturing sequential dependencies in NLP tasks. However, their inherent sequential processing limits efficiency and scalability. Transformers Vaswani et al. (2017) address these limitations by processing entire input sequences simultaneously through self-attention, enabling parallel computation and better handling of long-range dependencies compared to RNN-based models. State space models (SSMs) Gu et al. (2021) offer an alternative approach to sequence modeling by leveraging structured state representations and computationally efficient recurrence mechanisms. We consider the Associative Recurrent Memory Transformer (ARMT) Rodkin et al. (2024), an extension of the transformer designed to enhance memory capabilities. ARMT builds on the Recurrent Memory Transformer Bulatov et al. (2022) by incorporating quasi-linear attention mechanisms that improve information transfer across input blocks, mitigating limitations in long-context processing. We discuss the properties of these models in Appendix C.

We also explore several approaches for enhancing reasoning in neural networks, such as Chain-of-Thought, RL-methods (GRPO), and Adaptive Computations Time.

**Chain-of-Thought (CoT).** prompting Wei et al. (2022) is a technique for enhancing the reasoning capabilities of LLMs. Unlike standard prompting techniques, which attempt to directly infer an answer from the input, CoT forces the model to explicitly generate intermediate reasoning steps while solving a problem, allowing it to reference these tokens as a form of recurrent state. This mechanism effectively increases the formal computational power of the model Merrill & Sabharwal (2023a) and extends its effective depth enabling LLMs to perform multi-step reasoning, particularly in tasks such as mathematical problem-solving, logical inference, and commonsense reasoning Wei et al. (2022).

**Learning to reason with RL.** Another common practice involves training LLMs with reinforcement learning methods such as proximal policy optimization (PPO) Schulman et al. (2017) and group relative policy optimization (GRPO) Shao et al. (2024) after supervised finetuning in order to improve the generation of reasoning traces. RL post-training has been shown to improve instruction following Ouyang et al. (2022) as well as mathematical Wang et al. (2024) and general reasoning performance in LLMs Havrilla et al. (2024); Kumar et al. (2024); Guo et al. (2025). Compared to supervised methods, training to reason with GRPO requires no supervision on intermediate reasoning steps. It only relies on rewards from correct final answers and maintaining the desired format.

**Adaptive Computation Time (ACT).** (Graves, 2016) is the mechanism proposed to allow recurrent and self-attentive models to perform a variable number of computation steps within each time-step dynamically. The core idea is to enable different parts of the sequence to have different computational complexities, which is particularly useful for tasks with non-uniform requirements for computation. In this class of models a halting unit dynamically decides how much "thinking time" should take place at each step, thus adaptively scaling the effective reasoning depth of the model. For mathematical formulation check the Appendix E.

**Recurrent Memory Transformers**. As a trade-off between expressive recurrent models and efficiently trainable transformers, the Recurrent Memory Transformer was proposed (Bulatov et al., 2022). It leverages recurrent steps between the fixed-sized segments, while the tokens inside these segments are processed in parallel with the transformer model, which RMT augments. In the original RMT (Bulatov et al., 2022), the recurrent steps are performed by passing the output of special memory tokens from one segment to the input of the next segment. In the enhanced version of RMT: Associative Recurrent Memory Transformer (Rodkin et al., 2024), the recurrent steps are performed with quasi-linear attention in each transformer layer. In this work, we use the ARMT as a representative of recurrent memory transformers.

## 3 EXPERIMENTS

We start our study with testing contemporary LLMs on a commonsense, natural-language task that is *formally equivalent* to our 1D cellular automata (1dCA) setup. The goal is to assess how well current models can (i) infer a simple logic rule from observations and (ii) chain that rule for multiple steps.

**LLMs performance on the *Handsup* game.** A group of friends sits around a table. In each round $n$, every friend $i$ has a binary state: up (hand raised) or down. The hidden rule has radius $r \in \{1, 2\}$: the state of friend $i$ at round $n$ depends only on the $(2r+1)$-tuple $\{i-r, \ldots, i, \ldots, i+r\}$ from round

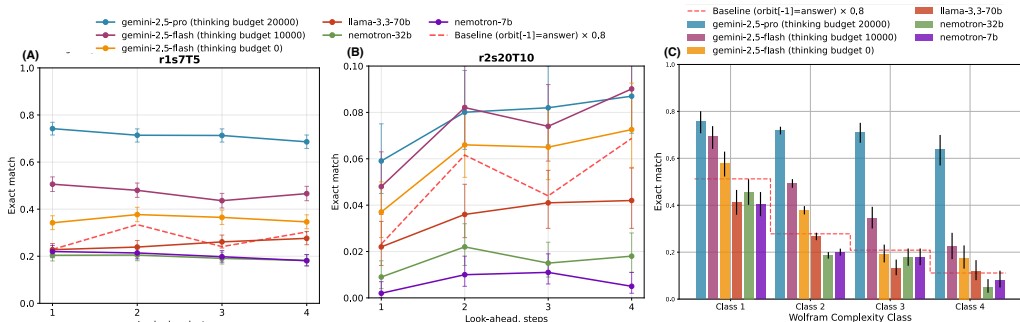

Figure 2: **Large Language Models struggle to solve reasoning 1dCA style tasks in natural language game.** **(a)** Only Gemini-2.5-pro sustainably achieves high scores in predicting the next state in Handsup game with $r=1$ and $w=7$ players given history of $T=5$ rounds. **(b)** None of models achieve reasonable performance (above 10%) for harder game with more distant dependencies ($r=2$), more friends (20) and longer history (10). Only Gemini models achieve scores higher then the *baseline* score, where the round is predicted the same as the last known round. **(c)** shows performance on the simple handsup game accross different subsets of the dataset, split with respect to the Stephen Wolfram's rule classification of ECA. We observe the performance degradation on the tasks with rules of higher complexity (Class 3 and 4). All values are mean exact match with $95\%$ CIs. The final game state was extracted from the models' answers with Gemma3-12B-IT model with $\approx 80\%$ EM extraction accuracy so we scale the baseline accordingly.

$n-1$. This is exactly an 1dCA-style local update $\rho : \{0,1\}^{2r+1} \to \{0,1\}$. We describe in natural language the first $N \in \{5, 10\}$ rounds and ask the model to predict the behaviour of players at round $N+s$ with $s \in \{1, 2, 3, 4\}$. We evaluate *exact-match* accuracy on the target round. To probe difficulty, we consider families of rules at $r=1$ and $r=2$, and for $r=1$ also group rules by Wolfram complexity class. We compare Gemini 2.5 Pro and Gemini 2.5 Flash with different "thinking budgets" (20k, 10k, and 0), Llama-3.3-70B, and Nemotron-32B/7B.

Figure 2 reports LLMs performance on Handsup game. In the simple game (fig. 2a) equivalent to elementary CA ($r=1, w=7, T=5$, 256 possible rules) only Gemini 2.5 Pro shows solid performance with a mild decline as look-ahead increases (about $0.72 \to 0.69$ from $s=1$ to $s=4$). Gemini 2.5 Flash performs lower with a thinking budget helps slightly over zero budget. Llama-3.3-70B and Nemotron 7B models hover around the trivial baseline across $s$. The hard game ($r=2, w=20, T=10, \approx 4.3B$ possible rules) represents a strong challenge to existing LLMs (see fig. 2b) as no model cross 10% EM with only Gemini models marginally over trivial baseline. As Fig. 2c shows accuracy decreases with dynamical complexity of the rule according to Wolfram's complexity classes. Gemini 2.5 Pro is consistently best (roughly $0.75$ in Class 1 down to $\approx 0.63$ in Class 4), Flash trails Pro, and open-weight baselines lag below $\approx 0.4$ in Classes 2–4. None of the models are uniformly robust across classes.

The *Handsup* results directly inform our research questions. The failure of most LLMs—even with "thinking" budgets—to solve the simplest radius-1 setting challenges the view that current successes reflect robust generalization rather than pattern recall (RQ1). Performance degrades systematically with more complex dependencies (r=2) and higher Wolfram classes, quantifying how difficulty scales with required reasoning steps (RQ2). Finally, to disentangle whether these failures stem from architectural limits versus training/inference procedures, we proceed to controlled small-model studies that test architectural capacity under matched supervision; success would implicate training as the bottleneck, while failure would argue for architectural changes (RQ3).

**Single-step performance across neural architectures.** We generated an 1dCA dataset with the CellPyLib Antunes (2021) for the fixed lattice size $W = 20$ and neighborhood radius $r = 2$. This configuration results in a total of $2^{2^{2r+1}} \approx 4.3 \times 10^9$ possible Boolean functions defining local rules. For each sample in the dataset, both the initial state and the local rule $\rho$ were generated randomly. We then computed the orbit for $T = 20$ time steps using these parameters. The training dataset consists of $9.5 \times 10^5$ instances and the test of $10^5$ instances. Importantly, the local rules included in the test set

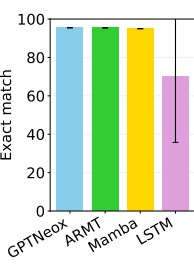
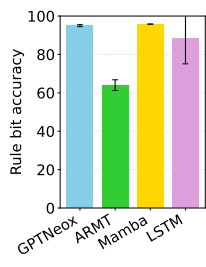
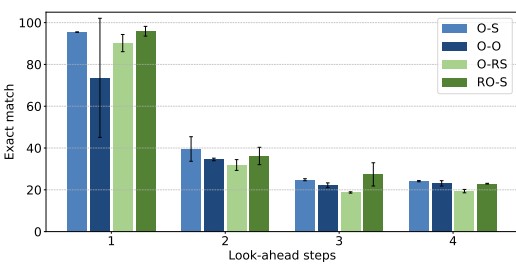

(a) State prediction     (b) Rule prediction     (c) GPTNeox on O-S, O-O, O-RS, RO-S tasks

Figure 3: **Single-step accuracy is near-perfect across models, but multi-step performance collapses. (a)** Exact-match accuracy for single-step *state prediction* (O-S): all models except LSTM achieve >95 %. **(b)** Bit-wise accuracy for *rule inference* (O-RS): most architectures recover the hidden Boolean rule, yet ARMT trails the rest. **(c)** GPTNeox accuracy on variable-horizon prediction across the four task variants (O-S, O-O, O-RS, RO-S): accuracy falls steeply with look-ahead $k$.

are exclusive and not present in the training set. This separation ensures that the model's performance reflects its ability to generalize to unseen rules, rather than simply memorizing the training data.

As shown in Figure 3(a), models with different architectures can predict one step forward with nearly perfect accuracy. LSTM performs slightly worse than other architectures, likely due to challenges in effectively encoding the binary state representation. Successful learning demonstrate that the Transformer model is capable of generalizing not only over initial conditions for a particular function — commonly the focus in studies of transformer trainability in CA domain — but also across different Boolean functions of fixed arity (5 in our case).

When tasked with predicting both future states and the underlying rules (O-SR setting), Figure 3(b) shows that models generally achieve high accuracy on rule prediction, though with interesting variations. ARMT notably struggles with accurate rule inference compared to other architectures, despite handling next-state prediction well.

**Limitations in the Reasoning Depth of Transformers.** We selected a 4-layer architecture with $d_{\text{model}}$=128 as a baseline configuration for our experiments. Using this configuration, we separately trained from scratch for each look-ahead step $k \in \{2, 3, 4\}$ of the O-S task the GPTNeox (Black et al., 2022) model to predict the state at time $x^{(T+k)}$ given an orbit $\mathcal{O}^T(x) = [x^{(1)}, x^{(2)}, \ldots, x^{(T)}]$. As presented in Figure 3(c), this task proved to be challenging. While the average accuracy for next-state prediction (O-S task with $k = 1$) was 0.95, it dropped to 0.40 for $k = 2$ and fell below 0.25 for $k = 3$ and $k = 4$. Despite having four layers, which in principle could capture up to two or three sequential transformations if effectively utilized, the model still struggles to learn look-ahead tasks for $k \geq 2$. Specifically, the same model's depth that suffices for the single-step O-S task is no longer adequate for maintaining accurate multi-step predictions, suggesting that the capacity is being taxed by the need to encode and apply repeated rule updates in a fixed number of transformations.

To determine whether this decline was due to the GPTNeox's architecture or the training objective, we explored whether accuracy could be improved by training the model to predict intermediate steps. This approach is analogous to multi-token prediction (Gloeckle et al., 2024). We employed the Orbit-Orbit (O-O) task, training the model to predict the next four states in parallel. The results, also shown in Figure 3(c), indicate that the model's predictive abilities degrade in this training scenario, as even prediction of the next state ($k = 1$) is less than 0.80 accuracy. However, the higher standard deviation suggests that it happens because of the instability of such training: some runs could simply fail, while others could work well (as shown by the exact match of $k = 2, 3, 4$ being relatively close to the O-S scenario).

These results suggest that learning to store a hidden representation of intermediate states (as in the O-S, O-RS and RO-S with $k > 1$) is hard for the model. Surprisingly, a direct supervision for a hidden representation of the underlying rule (O-RS) is more challenging initially and does not facilitate better generalization to longer planning horizons. This implies that explicitly encouraging

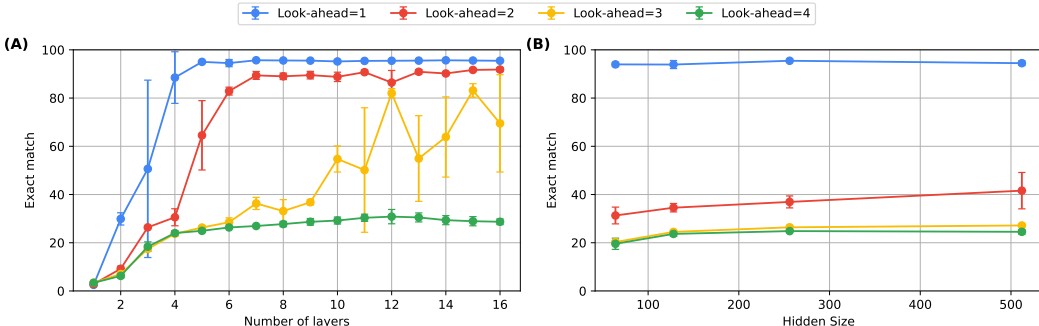

Figure 4: **Depth — not width — drives multi-step accuracy.** Exact-match accuracy for look-ahead horizons $k \in \{1, 2, 3, 4\}$ as a function of **(a)** transformer layer count and **(b)** embedding dimension $d_{\mathrm{model}}$. Deeper networks boost performance sharply for $k{\geq}2$ and plateau beyond six layers, whereas widening the model yields only marginal gains across all horizons.

the model to infer the generating rule cannot enhance its ability to make longer-term predictions by reinforcing the internalization of the system's dynamics.

Finally, we explored the scenario where the local rule $\rho$ is explicitly provided to the model, corresponding to the Rule and Orbit-State (RO-S) task. Intuitively, this should be the easiest task for the model, as it eliminates the need to infer the rule from the orbit. As shown in Figure 3(c), GPTNeox indeed learns to apply the given rule for next-state prediction with near-perfect accuracy for $k = 1$. Surprisingly, however, the performance for look-ahead steps $k = 2, 3$ and $4$ drops to the level of original O-S predictions. The poor performance on look-ahead steps $k > 1$ raises the question of whether this limitation stems from the neural network's parameter count, layer width, or width of its embeddings. To answer this question, we performed the experiments, while varying the number of transformer layers and the embedding dimension $d_{\mathrm{model}}$.

Figure 4 (a) shows that accuracy for one- and two-step prediction saturates after 4–6 layers. Three-step prediction, however, continues to improve up to about 12 layers, whereas four-step prediction remains poor regardless of depth. Figure 4 (b) examines width. Increasing $d_{\mathrm{model}}$ provides only marginal gains across all horizons, with the most noticeable bump occurring between 64 and 128 dimensions; further widening yields diminishing returns. These results illustrate the importance of increasing the model's depth rather than the width of its embeddings for better multi-step reasoning performance.

**Extending the depth of reasoning with Adaptive Computation Time.** The previous subsection confirmed that simply *adding layers* offers a clear performance boost, yet even a 12-layer transformer still falters for $k \geq 4$ (Fig. 4a). Here, we set the depth to 4 layers and study if it's possible to improve performance by techniques that expand a model's *effective* depth at inference time—segment-level recurrence and *Adaptive Computation Time* (ACT). Hyperparameters for all models can be found in Table 1. Both approaches inject extra computational steps without further increasing the static layer count, potentially enabling deeper reasoning while preserving parameter efficiency.

Figure 5 (A) shows that the auto-regressive models – GPTNeox, LSTM, and Mamba [2] – handle next-state prediction but fail to solve the multi-step task. Only ARMT manages to extend its capacity up to two look-ahead steps, likely because it processes sequences segment by segment and is thus forced to separate rule and state representations. This separation may enable the generation of a hidden representation for the intermediate state, followed by the application of the rule, effectively enhancing the depth of the model reasoning.

Augmenting models with ACT has little effect on all architectures except GPTNeox, which sees improved performance at $k = 2$ but not at $k = 3, 4$. Overall, ARMT makes effective use of

---

[2]We use the architecture from the previous section: 4 layer GPTNeox with $d_{\mathrm{model}} = 128$ and 4 attention heads. For Mamba, we use a state size of 16. For ARMT, $d_{\mathrm{mem}} = 32$. As ARMT is a segment-level model, we segment our state sequence in the way that each segment contains a pair of consecutive states in the orbit, and the prediction is performed in the last segment with the last CA state from the input in it. We report average results of 3 models trained with different seeds.

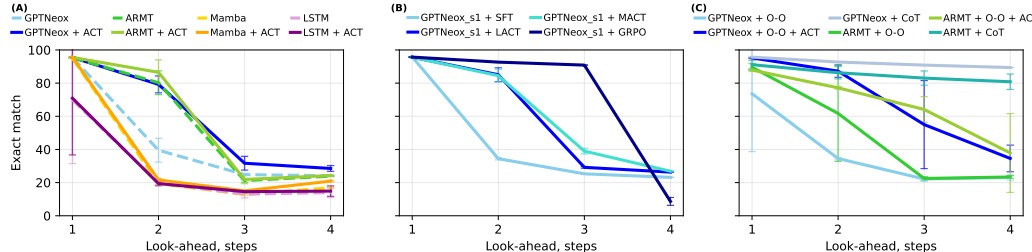

Figure 5: **Extensions of computation depth enhance the reasoning abilities of transformer-based models.** Values are exact match of the $x^{(T+k)}$ state prediction for look-ahead steps $k \in \{1, 2, 3, 4\}$. **(a)** ACT significantly improves computational abilities of transformer-based models in multi-step prediction. **(b)** Without supervision on intermediate reasoning steps RL training with GRPO allows the model to extrapolate reasoning on 3 steps forward. **(c)** With step-by-step supervision, the CoT approach significantly outperforms the in-depth approach of ACT. GPTNeox and ARMT with both ACT and O-O supervision perform the best.

the transformer's four-layer depth but cannot extend beyond it. Likewise, while ACT helps the transformer make use of its existing layers more efficiently, it fails to enable any architecture to solve three- or four-step predictions. Moreover, LSTM and Mamba are unable to master multi-step tasks with or without ACT, likely due to representation bottlenecks in their hidden states.

We subsequently chose to train GPTNeox model that is already capable of performing one-step reasoning with the SFT, LACT, MACT, and GRPO methods, with the goal of enabling it to reason over multiple steps without access to supervision for the intermediate reasoning stages. As illustrated in Figure 5 (B), standard supervised fine-tuning (SFT) fails to address the problem effectively. Although the model is primarily trained on a one-step prediction task, it struggles to apply the rule iteratively. Consistent with previous results (Fig. 5 (A)), applying ACT both at the layer level (LACT) and across the entire model (MACT) improves performance on the two-step prediction task but does not generalize beyond that. Interestingly, when trained using RL (GRPO) and granted the capability to autoregressively generate intermediate "thinking" tokens before producing the final output, the model succeeds on the three-step prediction task. The reward signal is defined as the average token-level accuracy of the model's prediction following the end-of-thinking token.

**Reasoning Supervision.** We examine the impact of reasoning supervision on GPTNeox and ARMT, along with their corresponding ACT-augmented variants. To this end, we replicate the O-O training setup by incorporating mask tokens into the autoregressive models within a causal masking framework. Figure 5 (C) shows that contrary to our expectations, the O-O training objective alone does not yield performance improvements for either GPTNeox or ARMT. However, the integration of O-O training with ACT results in superior performance, surpassing both the baseline and ACT-only variants.

As a final step, we combined GPTNeox and ARMT with a token-by-token CoT-like next-token prediction training. Under this regime, both models succeed at multi-step prediction up to $k = 4$, with GPTNeox slightly outperforming ARMT across each look-ahead distance (Fig.5 (C)). These results suggest that, when explicit reasoning supervision is available, a chain-of-thought-inspired approach to training offers a particularly effective strategy for enabling multi-step reasoning.

In addition to the cellular automata experiments, in Appendix H we show the significance of our findings on group multiplication benchmark (Merrill et al., 2024).

## 4 DISCUSSION AND CONCLUSIONS

Our study examines how *architecture*, *training signal*, and *depth-extension strategy* jointly determine a model's ability to learn multi-step reasoning in 1dCA—*without memorization*, since train/test rules are disjoint. The headline results (aggregated in Appendix A, Figure 6) speak directly to our research questions **RQ1–RQ3**.

- **Models can infer unseen rules, but LLMs falter on the simplest case (RQ1).** Both Transformers and recurrent/SSM variants (GPT-NeoX, LSTM, Mamba, ARMT) succeed on rule induction from orbits—evidence of genuine generalization because evaluation uses unseen rules. However, evaluated LLMs (except Gemini 2.5 Pro) fail to reliably solve even the radius-1 *Handsup* setting, indicating that scale and generic "think more" prompting are insufficient.
- **Reasoning difficulty grows sharply with look-ahead depth (RQ2).** Fixed-depth (4-layer) models $k{=}1$ but collapse for $k{\geq}2$, revealing a clear depth barrier.
- **Adaptive halting adds $\approx +1$ effective step at low compute cost (RQ3).** Adding Adaptive Computation Time (ACT) to a Transformer consistently shifts the depth frontier (roughly $k{:}1{\rightarrow}2$ or $2{\rightarrow}3$) without increasing parameters, with diminishing returns past $k{\approx}3$.
- **GRPO reaches three-step rollouts without intermediate supervision (RQ3).** RL rewarding final correctness only matches CoT@$k{=}2$ performance at $k{=}3$.
- **Token-level CoT saturates the current benchmark up to four steps (RQ3).** With stepwise targets, GPT-NeoX attains $> 99\%$ accuracy for $k{\leq}4$ (Figure 5C), showing that explicit supervision can elicit deeper computation given availability of intermediate labels.
- **Depth limits align with capacity constraints and can be partially mitigated (RQ2/RQ3).** Models with shallow effective depth (e.g., $TC^0$-like limitations) require more layers to track longer computations; ACT partially alleviates this on harder state-tracking (e.g., group-multiplication) tasks but does not fully resolve the gap.

**Broader implications for LLM reasoning—and beyond.** Our results align with a growing body of evidence that *reasoning failures often stem from insufficient depth allocation and sparse optimisation signals*. For LLMs, this suggests that (i) **prompt engineering alone is unlikely to improve multi-step reasoning**: unless intermediate steps are reinforced—via CoT, search-augmented decoding, or RL-style self-critique—models tend to default to shallow heuristics; (ii) **adaptive-depth mechanisms are a promising scaling direction**: ACT-style halting, deployed token-wise or layer-wise, can allocate computation on demand to match the variable complexity of real queries; and (iii) **explicit intermediate representations remain the most reliable route** to multi-step generalisation via CoT.

Beyond language, the same principles apply to neural algorithmic reasoning, robotic planning, and scientific simulation: whenever the target task contains latent iterative structure, giving the network *room*—via dynamic recurrence, learned halting, or supervised scratch-pads—to run the hidden algorithm is more data-efficient than brute-force depth. We therefore advocate future benchmarks that (a) separate rule induction from state propagation, (b) report *effective depth* alongside accuracy, and (c) evaluate adaptive-computation policies explicitly. Progress along these axes will benefit not only next-generation LLMs but also neural systems tasked with symbolic manipulation, formal verification, and open-ended planning.

**Conclusions.** We introduced 1dCA reasoning benchmark that isolates multi-step reasoning *without memorisation* by using disjoint train/test rule sets. Success therefore reflects genuine *rule inference* followed by iterative application, not lookup.

Empirically, fixed-depth (4-layer) models—Transformers, LSTMs, and state-space models—show a sharp depth cutoff: they solve $k{=}1$ but collapse for $k{\geq}2$. Segment-recurrent attention (ARMT) extends this to $k{=}2$ yet remains bounded. Adding Adaptive Computation Time (ACT) provides a compute-efficient $\sim +1$ effective step (with diminishing returns beyond $k{\approx}3$). Reinforcement learning via GRPO achieves reliable $k{=}3$ *without* intermediate labels, while token-level Chain-of-Thought attains near-perfect accuracy up to $k{=}4$. Complementing these small-model results, most contemporary LLMs—*except* Gemini 2.5 Pro—struggle even on the simplest natural-language proxy (radius-1 *Handsup*), underscoring that scale and generic "think more" prompting are insufficient.

Together, these findings support our contributions: (1) a benchmark that cleanly separates rule induction from state propagation; (2) a systematic architectural comparison; (3) an analysis of depth-extension mechanisms (recurrence, halting, RL, and explicit stepwise supervision); and (4) practical guidance on eliciting deeper computation. More broadly, they show that *how* we train and allocate compute can matter as much as *what* we train: objectives that force multi-step prediction and mechanisms that adaptively allocate depth are decisive, while explicit intermediate representations remain the most reliable route to deeper generalisation.

## LIMITATIONS

While our findings offer valuable insights into methods for enhancing reasoning, we acknowledge that the study is limited to small-scale models, and certain conclusions may not generalize directly to large language models. Our LLM evaluation covers only selected models over the main classes and sizes.

## REPRODUCIBILITY STATEMENT

Metrics are reported with 95% confidence intervals for handsup game with language models. In all small models finetuning experiments we report standard deviation estimates (square root of unbiased variance estimation) for confidence intervals. All hyperparameters are specified in Table 1, and we describe training details and used hardware in Section D. We also release the full codebase to ensure reproducibility of results: https://anonymous.4open.science/r/beyond_memorization.

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

## A  SUMMARY OF MODELS' PERFORMANCE

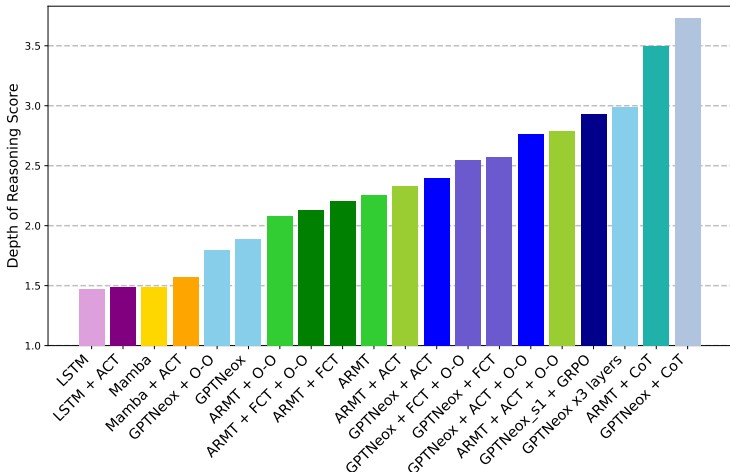

Figure 6: **With GRPO as well as with ACT and Orbit-Orbit training depth of reasoning can be significantly extended.** Average $DepthScore = 1 + \sum_{i=2}^{4} acc(i)$, where $acc(i)$ is the accuracy of predicting the $(10+i)$th state based on the first 10 states.

## B  RELATED WORK

**Computational Expressivity.** Sanford et al. (2024b) show that in setups where the input context length grows but the model depth remains constant, transformers achieve logarithmic complexity scaling in input size for sparse averaging tasks and linear scaling for triple detection. They further use the simulation of transformers in a constant number of MPC (Karloff et al., 2010) communication rounds to demonstrate their expressive power, showing that logarithmic-depth transformers can efficiently solve tasks that are intractable for graph neural networks and recurrent models Sanford et al. (2024a). Merrill & Sabharwal (2023b) prove that transformers with logarithmic precision can be simulated by constant-depth logspace-uniform threshold circuits, implying fundamental computational limitations. Zhang et al. (2024) employ circuit complexity theory to show that bounded-depth transformers cannot directly solve certain arithmetic or equation tasks, unless the model size increases exponentially.

**Formal Language Recognition.** The Chomsky hierarchy has been used to classify the computational capabilities of transformers and their expressivity limits. Deletang et al. (2023) show that transformers struggle with non-regular languages. Strobl et al. (2024) provide a comprehensive survey on how transformers relate to formal language classes, identifying the architectural constraints that limit their ability to process hierarchical structures. They show that while transformers with softmax attention can count, they remain within $\mathsf{TC}^0$ and struggle with evaluating Boolean formulas or solving complex hierarchical tasks. Zhang et al. (2024) discuss transformers' limitations due to their lack of recurrence, arguing that they are computationally weaker than recurrent models in formal language tasks.

Several studies explore how CoT enhances transformer reasoning capabilities. Feng et al. (2024) show that transformers can solve arithmetic and dynamic programming tasks via CoT, which they fail to do directly. Merrill & Sabharwal (2024) demonstrate that CoT increases computational power, enabling the recognition of regular languages. Nowak et al. (2024) formalize CoT reasoning probabilistically, showing equivalence to probabilistic Turing machines. Zhang et al. (2024) argue that CoT can approximate recurrent computation, mitigating transformers' lack of explicit recurrence.

There are generalizations of CoT that relax the human-like word-by-word out-loud reasoning. The reasoning process has been moved to special pause (Goyal et al., 2024), think (Herel & Mikolov, 2024), or filler (Pfau et al., 2024) tokens to allow the model to think internally before generating a response. Coconut (Chain of Continuous Thought) Hao et al. (2024a) further extends this by

replacing explicit word decoding with the model's last hidden state as input to the next step, effectively shifting reasoning into the latent space. Moreover, since real-world datasets rarely include supervision for long, multi-step reasoning, approaches that incorporate verifiers or intermediate feedback have become increasingly important (Pfau et al., 2024). At the same time, reinforcement learning methods (Schulman et al., 2017), such as GRPO (Shao et al., 2024), which rely solely on rewards for correct final answers, show great promise.

Overall, these studies highlight the limitations of transformers in reasoning depth and computational power, showing that CoT-like approaches and recurrence can help mitigate these constraints. Our work explores the use of One-dimensional Cellular Automata (1dCA) as a framework to evaluate models' reasoning abilities. 1dCA provides a flexible and controlled setting where the number of sequential steps required to solve a task can be precisely defined. Adjusting the complexity of state transition rules allows for varying task difficulty.

**Looped Transformers** Another paper (Yang et al., 2024) investigates whether looped transformers (Yang et al., 2024) can emulate iterative learning algorithms, such as gradient descent, for data-fitting problems like linear regression. Their core finding is that looped transformers can achieve comparable performance to standard transformers with significantly fewer parameters by effectively replicating these iterative optimization steps. Our paper investigates how different architectures and training methods affect a model's ability to learn and perform multi-step reasoning and rule abstraction. The "iterations" in our study are interpreted as steps for applying a discovered rule or propagating a state, which is distinct from emulating optimization algorithms.

RELAY (Yu et al., 2025) is a framework that aligns CoT steps with loop iterations and uses intermediate supervision during looped transformer training to generate high-quality reasoning chains for auto-regressive models. Their aim is to leverage the length generalization of looped transformers to improve auto-regressive models' handling of longer reasoning chains. In our paper, we study CoT as a training objective that provides direct reasoning supervision on intermediate states for multi-step state prediction on 1dCA. While both studies involve recurrence and CoT-like supervision, Yu et al.'s work focuses on a specific methodology for generating CoT for other models by aligning CoT steps with loops, whereas our work directly evaluates how training with or without intermediate supervision, as in O-O or GRPO, respectively, influences a model's core reasoning capabilities in a disentangled environment.

In the "Illusion of Thinking" research (Shojaee et al., 2025) authors show that the models' performance decreases with the increased complexity of puzzle environments. For thinking models, however, this degradation is less dramatic. Which is consistent with our findings on Figure 5 (A).

## C MODELS DISCUSSION

**LSTM** By integrating a gating mechanism into recurrent neural networks, LSTMs alleviated the vanishing gradient problem, allowing the model to retain information from up to 10–15 prior time steps. However, LSTMs still face several limitations. First, despite the gating mechanism, they often struggle with very long-range dependencies, as information can decay over extended sequences. Second, their sequential nature hinders parallelization, which slows training and increases computational cost compared to more modern architectures such as transformers. As a result, while LSTMs represented a major breakthrough in sequence modeling and in theory can process contexts of infinite length, they have been largely superseded by more scalable and efficient models.

**Transformers** Attention mechanism allows transformers to dynamically focus on relevant parts of the input, facilitating effective information integration across long distances. As a result, they maintain and reuse context more effectively than LSTMs, making them a powerful backbone for modern large language models. This design has enabled state-of-the-art performance on complex reasoning tasks, cementing the transformer's role at the forefront of natural language processing.

While this flexibility is powerful, it also introduces drawbacks. Transformers must compute and store a large attention matrix, often scaling to $O(n^2)$ in both memory and computation. This creates challenges when handling very long inputs or generating lengthy outputs, as hardware and software limitations cap the practical context window. Another limitation of transformers is their difficulty in processing information "in-depth." Each generation step requires a fixed amount of computation,

| Model | Depth | $d_{\text{model}}$ | $d_{\text{mem}}$ / state_size | $n_{\text{heads}}$ | max ACT iterations |
|---|---|---|---|---|---|
| GPTNeox | 4 | 128 | - | 4 | 4 |
| ARMT | 4 | 128 | 32 | 4 | 4 |
| Mamba | 4 | 128 | 16 | - | 4 |
| LSTM | 4 | 128 | - | - | 4 |

Table 1: **Hyperparameters for the base models.** We used these hyperparameters in the O-S, O-O, O-RS and RO-S experiments, as well as CoT and GRPO experiments.

constrained by the number of transformer layers. Consequently, transformers face challenges with multi-hop reasoning. To enable more efficient in-depth reasoning, various test-time compute strategies have been introduced, including chain-of-thought prompting, Monte Carlo Tree Search, and others. While these techniques partially mitigate the issue, they remain bottlenecks: longer generations demand substantial computational resources and may exceed the effective context window. These techniques also require supervision for intermediate steps to train the model. This is a huge limitation as strong AGI systems should automatically learn to recursively apply rules to data.

**State Space Models** While less prevalent compared to RNNs and transformers, SSMs are widely used in control theory and signal processing. In the context of neural networks, SSMs aim to combine the strengths of recurrent models, such as handling infinitely large contexts, with the efficiency of convolutional models for fast prompt processing and training. This positions SSMs as a middle ground between classical LSTMs and transformers.

In our experiments, we utilize Mamba, an SSM variant improved with a selective mechanism (Gu & Dao, 2023; Gu et al., 2021). The Mamba Selective State Model extends this framework by making $A$, $B$, and $C$ dynamic, adjusting them based on the input $x(t)$. This adaptive mechanism allows Mamba to selectively focus on relevant input features, filtering out irrelevant details (Gu & Dao, 2023). By dynamically adapting its parameters, Mamba is able to capture long-range dependencies in sequences while remaining computationally efficient.

While SSMs excel in efficiently modeling long-range dependencies and processing sequential data with reduced computational overhead compared to transformers, they typically lack the expressiveness and flexibility required for advanced reasoning tasks. These models may face challenges in capturing complex, hierarchical relationships, compounding the limitations already present in transformers when it comes to in-depth reasoning.

**Associative Recurrent Memory Transformer** As shown in Rodkin et al. (2024), ARMT can leverage information from the distant past of up to 50 million tokens. Compared to SSMs, ARMT is more expressive due to its grounding in the classical transformer architecture, while it also introduces the ability to recurrently process contexts of infinite length.

**Theoretical Depth Estimates** Theoretical estimates predict that for GPTNeox and Mamba depth of computation is limited by the number of layers $Depth = O(L)$, where $L$ is the number of model layers. For LSTM computational depth not only grows with the number of layers, but also with the sequence length, making $Depth = O(L + N)$, here $N$ is the sequence length. ARMT is a trade-off between parallelization and recurrence. It utilizes the forward transformer for local processing of the segment, but passes its recurrent state between segments in RNN-like format, which allows its computational depth to grow with the sequence length, making $Depth = O(L + \frac{N}{S})$, here $S$ is the segment size.

# D  TRAINING DETAILS

We train all our models for 40k steps with Adam optimizer with learning rate 3e-4 with linear warmup for 1000 steps and linear decay. We use total batch size of 256 samples. The vast majority of experiments we ran on single NVIDIA RTX 6000 Ada GPU. Models hyperparameters can be found in Table 1.

# E   ADAPTIVE COMPUTATION TIME FORMULATION

The module calculates a halting weight $p_t$ at each computation step $t$, which represents the percentage of the task completed by the module $f$:

$$p_t = \text{HALT}(h_t); \quad h_{t+1} = f(h_t), \quad \text{HALT}(h_t) = \sigma(W_h h_t + b_h) \tag{1}$$

where $h_t$ is the layer input. This weight is accumulated into $P_t$ until the halting condition is met:

$$P_t = \sum_{i=0}^{t} p_i; \quad T = \text{argmin}_t(P_t \geq 1 - \epsilon) + 1. \tag{2}$$

Finally, the prediction is done in the following way: $y = \sum_{t=0}^{T-1} p_t h_{t+1}$ with $p_{T-1} = R = 1 - \sum_{t=0}^{T-2} p_t$. For training, we add an auxiliary component to the loss function $\hat{L} = L + \tau R$. This component serves as a time penalty.

# F   SAMPLES EXAMPLES

## F.1   HANDSUP GAME

```
You peek through a doorway into a cosy room.
7 friends sit around a round table in this order: Alice, Bob, Carol,
Dave, Erin, Frank, and Grace - and then back to Alice again.
They don't talk. At the end of each round they all decide, at the very
same moment, either to raise a hand or to keep both hands on the table.

You watch and jot down what happens:
- Round 1. Alice, Bob, Dave, Erin, Frank, and Grace raise their hands.
    The others keep their hands on the table.- Round 2. Alice, Carol,
    Erin, Frank, and Grace raise their hands. The others keep their hands
     on the table.- Round 3. Bob, Dave, Frank, and Grace raise their
    hands. The others keep their hands on the table.- Round 4. Alice,
    Carol, and Erin raise their hands. The others keep their hands on the
     table.- Round 5. Bob and Dave raise their hands. The others keep
    their hands on the table.
Now it's your turn to be the clever observer.
Puzzle: What will each friend do in Round 6?
Please answer in plain words, going in order around the table, starting
    from the first name above. Answer with the list of people with hands
    up, not mentioning the ones with hands down. For example: Alice, Bob,
     and Dave raise their hands.
```

## F.2   ECA - R2S20T10

The samples from our open dataset.

The input vocabulary of the tested models consists of the following tokens: [0], [1], and [SEP]. The states and the local rule $\rho$ are encoded as binary strings. The model receives the orbit as a sequence of bits, representing consecutive states separated by the [SEP] tokens.

We train the model to predict the blue tokens.

In all these examples rule is 01011111100100000101111011111100 and the initial state is 10110111001000110100.

### O-S

```
10110111001000110100<sep>11101001101111101100<sep>10111011010000111011<sep>
11001110111011101100<sep>10111011001100111011<sep>11001110111011101100<sep>
10111011001100111011<sep>11001110111011101100<sep>10111011001100111011<sep>
11001110111011101100<gen>10111011001100111011
```

### O-O

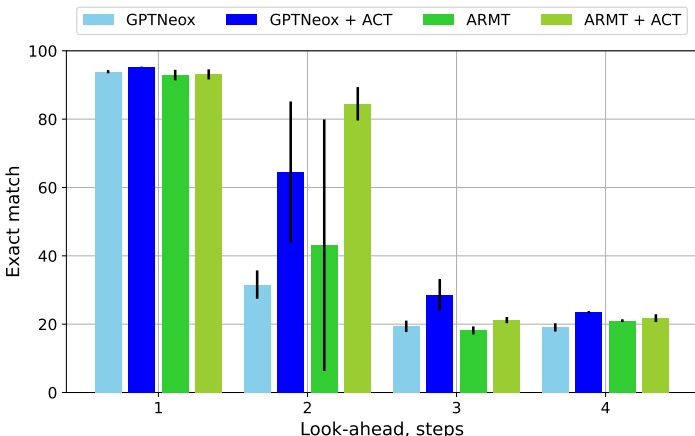

Figure 7: **ACT outperforms the base model on multiple prediction horizons task**. Exact match accuracy (mean $\pm$ std) for cellular automata state prediction across different look-ahead horizons. Models receive initial 10 states followed by a special shift token (1-4) indicating prediction horizon.

```
10110111001000110100<sep>11101001101111101100<sep>10111011010000111011<sep>
11001110111011101100<sep>10111011001100111011<sep>11001110111011101100<sep>
10111011001100111011<sep>11001110111011101100<sep>10111011001100111011<sep>
11001110111011101100<gen>10111011001100111011<sep>11001110111011101100<sep>
10111011001100111011<sep>11001110111011101100
```

**O-RS**

```
10110111001000110100<sep>11101001101111101100<sep>10111011010000111011<sep>
11001110111011101100<sep>10111011001100111011<sep>11001110111011101100<sep>
10111011001100111011<sep>11001110111011101100<sep>10111011001100111011<sep>
11001110111011101100<gen>01011111100100000101111011111100<sep>
10111011001100111011
```

**RO-S**

```
01011111100100000101111011111100<sep>10110111001000110100<sep>
11101001101111101100<sep>10111011010000111011<sep>11001110111011101100<sep>
10111011001100111011<sep>11001110111011101100<sep>10111011001100111011<sep>
11001110111011101100<sep>10111011001100111011<sep>11001110111011101100<gen>
10111011001100111011
```

## G   MULTIPLE PREDICTION HORIZONS TRAINING

Given an orbit $\mathcal{O}^T(x)$ and the random shift token $s_i \in \{s_1, s_2, s_3, s_4\}$ the objective is to predict the state $x^{(T+i-1)}$. In this setup, we train the model to reason more for some inputs than others.

We conducted experiments where a single model was trained to handle multiple prediction horizons (1-4 steps ahead) using special shift tokens in the input format: `[x_0][SEP]...[x_9][shift_k][gen][MASK]` where $k \in \{1, 2, 3, 4\}$ indicates the required look-ahead. As shown in Figure 7, baseline GPTNeox performs 32% shift=2 and 19% for shift=4. Introducing ACT substantially mitigates these drops.

The ARMT architecture shows comparable characteristics – while baseline performance at shift=2 is stronger than GPTNeox (43% vs 32%), ACT provides similar absolute improvements (85% at shift=2). However, both architectures exhibit similar limitations at the longest horizons (shift=4), with all variants scoring 21%-25%, indicating challenges in extreme-depth reasoning.

## H    GROUP MULTIPLICATION TASK

The task is, given the sequence of elements of some group, label each element with the product of all previous elements of the sequence, including the current one. This task is relevant to reasoning because it provides a controlled setup with tasks of different computational complexity.

We evaluated our models in 3 groups of different difficulty: $Z_{60}$, $A_4 \times Z_5$, and $A_5$; and different sequence lengths: 5, 10, 15, 20, and 40. For each model, we report the minimal number of layers to achieve 70% exact match accuracy. For the sake of consistency with previous works, we slightly changed the hyperparameters of our models. We use $d_{\text{model}} = 512$ and $n_{heads} = 8$. For the ARMT model, we use the segments of size 2.

As shown in Figure 8, the required depth for solving longer tasks grows for GPTNeox and Mamba models, while staying constant (1-2 layers) for the models with recurrence (ARMT and LSTM). Moreover, depth requirements can be significantly reduced by adding Adaptive Computation Time (ACT) or Associative Memory (ARMT), which is consistent with our findings on the 1dCA benchmark. LSTM, however, performs much better, being able to solve the problem with just one layer.

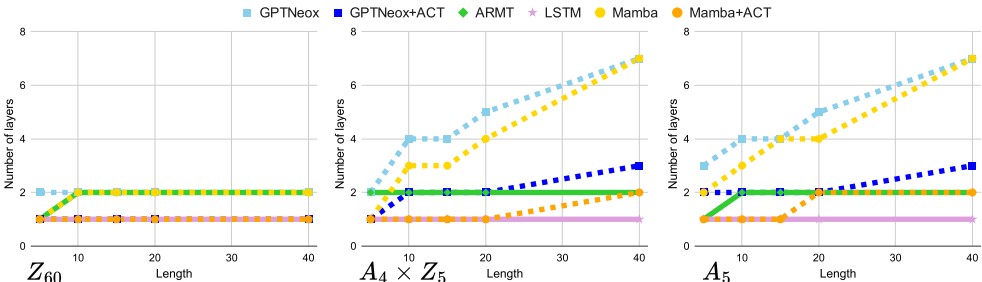

Figure 8: **ACT significantly reduces the required models' depth for the majority of group multiplication tasks.** Each chart contains the information about the minimal required number of layers for solving task of given length with 70% exact match accuracy. GPTNeox and Mamba being $TC^0$-limited models require more layers for solving deeper (longer in this case) tasks, while ARMT and LSTM solve them with constant number of layers.

## I    ABLATION STUDIES

Originally, ACT was applied to single-layer NNs (Dehghani et al., 2019; Graves, 2016). When it comes to deep models, we can apply ACT to each layer of the model, averaging the remainders over the layers to add as the time penalty to the loss (layer-wise ACT or LACT). Another option is to apply ACT to the whole backbone model (MACT), which maps the $\mathbb{R}^{N \times d} \to \mathbb{R}^{N \times d}$ (therefore without embedding and unembedding layer). In our ablation studies, we compare layer-wise ACT and model ACT but find that they perform similarly. See I.2 for more details. Therefore, in the main experiments, we use only layer-wise ACT and always refer to this version.

To determine whether performance gains stem from the adaptive nature of computation time or merely from increased computation, we include a fixed computation time (FCT) baseline in our ablation study (I.1). Specifically, we examine the case of three fixed iterations, chosen to match the upper bound of the average number of ACT operations observed in our experiments.

Here, we present several auxiliary studies of various ACT variants.

### I.1    FIXED NUMBER OF STEPS IN ACT VS DYNAMIC NUMBER OF STEPS

We conduct experiments with a fixed number of steps to assess the need for adaptivity in computation time. A constant depth of 3 was selected based on experiments with ACT, which demonstrated that this represents the upper limit of the number of steps reached for any hidden state. The results with Fixed Computation Time (FCT) and ACT as the baseline are presented in Figure 9 and Figure 10 for O-S and O-O settings respectively.

In O-S setting, FCT improved the exact match in look-ahead 2, 3 for GPTNeox, but performed worse in look-ahead 2 for ARMT. In contrast, in the O-O setting, FCT showed reduced performance for both GPTNeox and ARMT in look-ahead 2, 3, 4.

Therefore, adaptivity in computation time might find the optimal amount of steps leading to enhanced exact match, or perform equivalently with fewer steps.

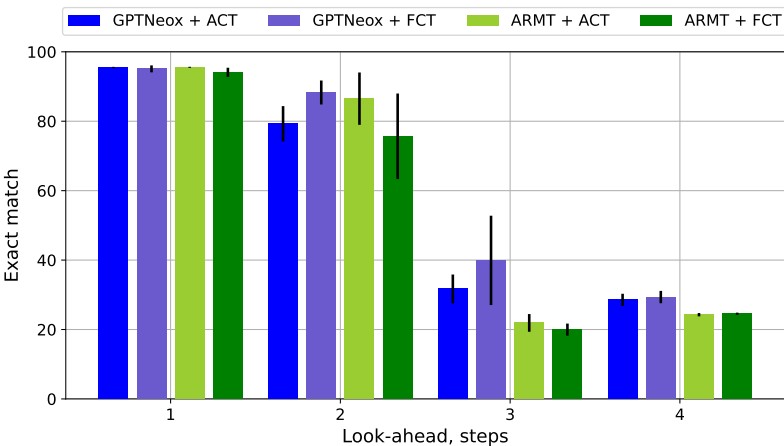

Figure 9: **Fixed Computation Time (FCT) with 3 iteration steps performs on par with Adaptive Computation Time (ACT) in Orbit-State task.** Exact match accuracy (mean ± std) for cellular automata state prediction across different look-ahead horizons.

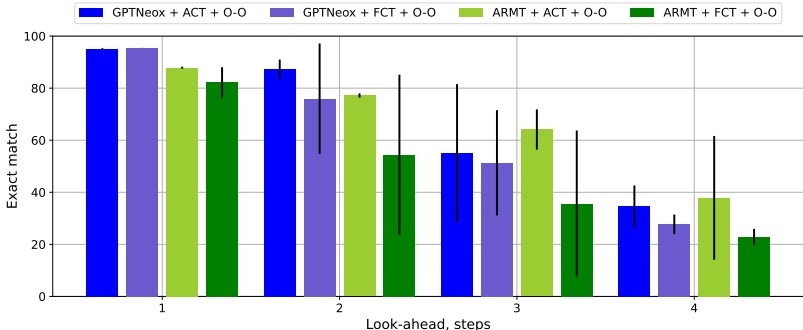

Figure 10: **Fixed Computation Time (FCT) with 3 iteration steps underperforms Adaptive Computation Time (ACT) in Orbit-Orbit task.** Exact match accuracy (mean ± std) for cellular automata state prediction across different look-ahead horizons.

## I.2 MODEL-ACT VS LAYER-ACT

Figure 11 shows that Layer-ACT performs similarly or better compared to Model-ACT. In particular, Model-ACT has a similar processing pattern to the COCONUT model (Hao et al., 2024b), passing the hidden states from the model output to the input. Therefore, a similar reasoning behavior is expected. A notable difference is observed when these types of ACT are applied to ARMT. However, it is important to note that training was stopped after 30,000 steps, and the model with MACT augmentation did not have sufficient time to fully converge. All models in this experiment adhered to these training restrictions to ensure a fair comparison.

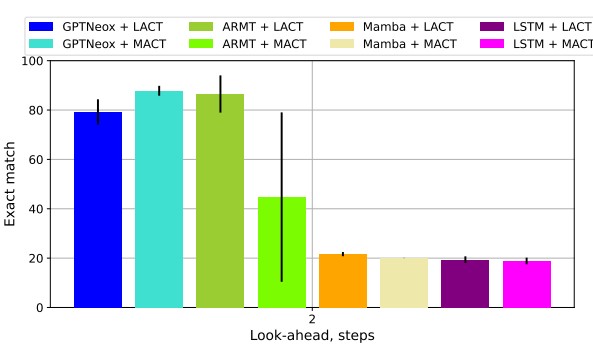

Figure 11: Layer-ACT performs similar or better compared to Model-ACT. Exact match on cellular automata state prediction task with look ahead 2.

