# OpenReview forum: "Beyond Memorization: Extending Reasoning Depth with Recurrence, Memory, and Test Time Compute Scaling"
_ICLR.cc/2026/Conference — ICLR 2026 Conference Withdrawn Submission_

### Official Review · Reviewer_uELE · 2025-10-16

**Soundness:** 2
**Presentation:** 3
**Contribution:** 1
**Rating:** 0
**Confidence:** 5

**Summary:**

The paper studies multi-step reasoning in neural networks using a controlled one-dimensional cellular automata ($1$dCA) benchmark that prevents memorization by using **disjoint train/test rule sets**. Models must (i) infer the hidden local rule and (ii) **chain** it for $k$ future steps. The authors (a) introduce four task variants (O-S, O-O, O-RS, RO-S) to tease apart rule induction vs. state propagation, (b) compare small Transformers, LSTMs, SSMs (Mamba-style), and a recurrent-memory Transformer (ARMT), and (c) test **depth-extension** strategies: Adaptive Computation Time (ACT), GRPO-style RL that induces "thinking tokens'', and token-level Chain-of-Thought (CoT) supervision. Main findings are that single-step prediction and rule recovery are easy, but accuracy **collapses with look-ahead ($k$)**; **depth helps more than width**; ACT gives about "$+1$ effective step''; ARMT reaches $k \approx 2$; GRPO reaches $k=3$ without intermediate labels; supervised token-level CoT attains near-perfect accuracy up to $k=4$. A natural-language proxy ("Handsup'' task) mirrors these trends for today's LLMs.

**Strengths:**

* **Clear problem decomposition.** The O-S / O-O / O-RS / RO-S variants cleanly separate rule induction from state propagation/chaining.
* **Memorization control.** Disjoint train/test rule sets are an effective way to force generalization beyond lookup.
* **Systematic comparisons.** Side-by-side evaluation of depth vs width, ACT vs GRPO vs CoT, and inclusion of a memory-augmented transformer (ARMT).

**Weaknesses:**

* **Novelty.** The main conclusions substantially overlap with prior works:

  * **Error compounding / exposure bias** and the failure of one-step training under rollout (Scheduled Sampling from Bengio et al. 2015; DAgger from Ross et al. 2011).
  * **Depth/recurrence/memory** as remedies for multi-step computation (End-to-End Memory Networks; Neural Turing Machine / DNC; Universal Transformer iteration).
  * **Test-time compute via reasoning tokens** (CoT from Wei et al. 2022; self-consistency) and **RL-induced scratchpads**.
  * **Neural CA literature**: learning CA rules and iterative local updates (Wulff & Hertz 1992; Mordvintsev et al. 2020 on Neural CA; follow-ups applying NCA to ARC and other CA/GCA tasks).
    The paper does not adequately acknowledge or distinguish itself from these lines.
* **Compute/accounting confounds.** Comparisons between **deeper nets**, **ACT micro-steps**, **GRPO**, and **token-level CoT** do not rigorously normalize for **effective compute** (parameters × steps × tokens). Some gains may stem from simply doing more computation rather than better mechanisms.
* **Limited ablations on rollout stability.** Known fixes for exposure bias (scheduled sampling, professor forcing, data aggregation) are not evaluated. These could narrow the gap between one-step training and multi-step rollout without the need for CoT or RL.
* **Architectural scope.** Depth vs width conclusions are drawn for a specific small Transformer. Missing baselines include **Universal Transformers** (iterative layers), **RWKV/parallel RNNs**, or **explicit algorithmic modules** (e.g., learned rule tables), which are directly relevant in CA settings.
* **Theory gap.** The compounding-error narrative is intuitive but not formalized (e.g., Lipschitz constants or light-cone arguments for CA). A lightweight analysis could sharpen claims and predict the observed breakpoints.

**Questions:**

1. **Compute normalization:** How do you normalize **effective compute** across methods (deeper layers vs ACT halting steps vs RL-generated thinking tokens vs CoT length)? Can you report accuracy as a function of *total FLOPs* and *latency*?
2. **Exposure-bias baselines:** What happens if you add **scheduled sampling** or **DAgger-style** data aggregation to the one-step training? Does this close some of the multi-step gap without CoT or RL?
3. **Universal Transformer / iterative layers:** Have you tried an **iterated (weight-tied) transformer** to separate “depth” from parameter count? This seems directly targeted at iterative computation without blowing up parameters.
4. **Why ARMT underperforms on rule inference?** You report ARMT trailing others on O-RS bit accuracy. Is this due to segment design, memory contention, or attention pattern? Any ablation on segment length or memory size?
5. **Rollout training objectives:** Your O-O supervision did not help much. Did you try **teacher-forcing vs free-run** mixtures, **curricula** on k, or **intermediate loss shaping** (e.g., consistency losses) to stabilize multi-step training?
6. **Theoretical framing:** Can you bound the expected **Hamming error growth** under your models’ Lipschitz constants, or use CA light-cone arguments to justify the linear/exponential regimes you observe?
7. **LLM proxy:** In the Handsup task, could **self-consistency** or **verification-guided decoding** close the gap for open-weight models, or is the failure robust even with these stronger test-time procedures?

**Details Of Ethics Concerns:**

No concerns.

---

> ### Author Response · Authors · 2025-11-21
> **Rebuttal to the Reviewer uELE**
>
> We sincerely thank the reviewer for the comprehensive and detailed comments. We appreciate the effort taken to analyze our work in depth. Below we clarify the unique contributions of our paper, address the raised concerns, and provide additional evidence and context.
>
> We agree that many cited works have discussed depth, recurrence, and exposure bias. Our contribution is not a repetition of those findings, but a unified, controlled, and quantifiable framework (1dCA) that allows isolating reasoning depth, recurrence, and test-time compute under **identical architecture and training conditions**.
>
> Below, we present the contributions that extend beyond those discussed in the aforementioned works:
> - Prior works use heterogeneous benchmarks (language, arithmetic, symbolic reasoning), making it difficult to disentangle memorization from reasoning.
>
> - Our 1dCA design (disjoint train/test rule sets and symbolic cellular transitions) focuses on the pure reasoning, making the model to infer the rule from the underlying sequence and apply it several times.
>
> - Using this framework we specifically show how the bounded nature of the transformer model can be extended using the ACT, CoT (i.e. autoregressive supervised reasoning trace generation in our case), or Recurrent Memory (ARMT), and to what extend this extension works (ACT= +1 effective step from 4 in the original transformer = ~100% reasoning depth increase = ARMT-augmentation; GRPO with intermediate reasoning = +2 effective steps = ~200% reasoning depth increase, CoT = at least + 3 effective steps = at least +300% reasoning depth increase)
>
> 1. **Compute normalization.** The work did not focus on compute optimization, that’s why we didn’t restrict the models with the amount of computation they did. Instead, we observed the primary limitations of the architectures, which was the goal of this work.
>
> 2. **Exposure-bias baselines.** We appreciate the suggestion to include scheduled sampling or DAgger-style aggregation. While we did not include these variants in the current submission, we agree they represent a valuable future direction to further isolate the role of supervision versus architectural bias in multi-step generalization.
>
> 3. **Universal Transformer / iterative layers.** We thank the reviewer for pointing out universal transformers. Our GPTNeoX+ACT configuration is, by definition, an instance of a multi-layer Universal Transformer, where adaptive halting introduces dynamic iteration depth. In fact, our implementation directly reuses the original Universal Transformer codebase, ensuring a consistent comparison to this class of iterative architectures. Note that in ACT iterative layers share their weights.
> Regarding the RWKV/parallel RNNs we actually conducted a few experiments with RWKV-6 and RWKV-7 and didn’t observe the significant difference from the existing SSMs representative in our work – Mamba, so we didn’t include these result in the final version of the paper.
>
> 4. **Why ARMT underperforms on rule inference.** The reviewer correctly notes that ARMT trails slightly on the O-RS condition. This appears linked to the segment-based memory layout, where rule-induction and state-prediction compete for limited memory state. We will clarify this discussion in the revision and note that such trade-offs are intrinsic to recurrent-memory transformers under fixed capacity.
>
> 5. **Rollout training objectives.** We actually tried teacher forcing in our CoT experiments and it worked well up to the k=4. Regarding the other methods, we didn’t try them but appreciate the suggestion and agree that incorporating mixed rollout objectives and intermediate supervision could further stabilize long-horizon training and plan to explore these directions in future work.
>
> 6. **Theoretical framing.** We appreciate the insightful suggestion regarding formal analysis. We agree this would  extend the interpretation of compounding error and intend to incorporate such framing in future work. Our experiments are, however, enough to prove the claims in our paper.
>
> 7. **LLM proxy.** We thank the reviewer for the suggestion to test additional decoding strategies such as self-consistency or verification-guided decoding. While not explored here, these are promising extensions to probe whether deeper inference steps can offset reasoning depth limitations. We, however, believe that the conducted experiments provide enough motivation for exploring reasoning beyond memorization.
>
> We hope these clarifications help convey the soundness and lasting value of our contribution.

---

> > ### Comment · Reviewer_uELE · 2025-11-24
> > **Final rating: unchanged**
> >
> > Thank you for the rebuttal.
> >
> > Most of my main concerns, especially around (i) novelty relative to prior work on exposure bias, iterative/algorithmic architectures, and neural CA and (ii) the lack of compute-normalized comparisons and (iii) the absence of key ablations (scheduled sampling/DAgger-style fixes, additional iterative baselines, and a more formal treatment of error growth), are not addressed by new experiments or analyses in the rebuttal. As a result, while I appreciate the additional context and clarifications, they do not substantially change my assessment of the strength and distinctiveness of the contribution in its current form.
> >
> > Accordingly, I will keep my overall score and confidence unchanged.

---

> > > ### Author Response · Authors · 2025-11-25
> > >
> > > We would kindly ask you to clarify which of the following claimed novelties you find unconvincing:
> > >
> > > - Introducing a new memorization-free framework for reasoning estimation.
> > >
> > > - A new Handsup task – a worded proxy equivalent to the 1dCA update – used to assess LLMs under varying look-ahead and rule complexity, showing that many LLMs (except Gemini 2.5 Pro) fail on the simplest radius-1 setting and all of them fail on the radius-2.
> > >
> > > - Providing a comprehensive comparison of architectures, demonstrating how the bounded nature of the Transformer model can be extended through ACT, CoT (i.e., autoregressive supervised reasoning trace generation in our case), or Recurrent Memory (ARMT). We further quantify the extent of these improvements:
> > >
> > >
> > >     + ACT = +1 effective reasoning step (from 4 in the original Transformer), corresponding to roughly a 100% increase in reasoning depth (ARMT-style augmentation);
> > >
> > >
> > >     + GRPO with intermediate reasoning = +2 effective steps (~200% increase in reasoning depth);
> > >
> > >
> > >     + CoT = at least +3 effective steps (~300% or more increase in reasoning depth).
> > >
> > >
> > > Regarding the compute-normalized comparisons, please note that the goal of this work is not to identify the most compute-efficient algorithm, but rather to analyze different approaches without constraining computational resources, in order to assess their peak achievable performance.
> > > Finally, regarding the baselines, we believe our study already provides a comprehensive set: four architectures, each combined with their ACT variants and multiple training procedures, resulting in a broad and detailed comparison.
> > >
> > > We appreciate your time and feedback. However, given that the rebuttal directly addresses the points raised -- both conceptually and experimentally -- we would be grateful if you could specify which aspects of our contributions or methodology you still find unconvincing. Your clarification would help us better understand and improve upon the concerns you have raised.

---

### Official Review · Reviewer_TX7H · 2025-10-26

**Soundness:** 3
**Presentation:** 3
**Contribution:** 2
**Rating:** 6
**Confidence:** 2

**Summary:**

This paper studies the multi-step reasoning ability of LLMs in a controlled cellular-automata (1dCA) framework. The objective is to let models learn hidden rules from disjoint train and test sets. The experiments reveal that increasing model depth is crucial, and extending effective depth via recurrence, memory, or test-time compute improves results but remains bounded. In particular, experiments show that 4-layer models struggle with 2-step reasoning tasks, and test-time computing methods such as Adaptive Computation Time (ACT) and Chain-of-Thought (CoT), and achieve 3 or 4 steps. In the discussions, authors raise the claim that their work adds to the evidence that reasoning failures often stem from insufficient depth allocation and sparse optimisation signals.

**Strengths:**

- Experiments are well-designed, novel (to my understanding).
- Model choices and ablation studies support the central claim of the paper.
- Easy to understand and well-organized writing flow.

**Weaknesses:**

- The work seems largely empirical, and the novelty of the claim is not much.

The relationship between model depth and reasoning ability has been studied extensively by previous works. The authors provided a very good related works section, but I don't see the central claim to be much different from the established belief in the community. To be exact, this work proposes a new method (1dCA) to study reasoning and model depth and achieves the same conclusion as previous works. The authors provide a contribution list at the end of the introduction, but it only highlights the new 1dCA method. While this is definitely helpful, especially with the comprehensive experiments on different test-time compute methods, I still struggle to see this work's broader significance in the community.

- Some results are a little bit concerning (see questions)

**Questions:**

1. Besides empirical methods, please try to also clarify the novelties in the understanding of test-time compute and reasoning models (if there are any).
2. How is CoT achieved in the model? I might have missed it, but I can't find details of CoT-related experiments.
3. k=4's result in Figure 4A is a little concerning. Under the current understanding that model depth has a strong relationship with reasoning ability, we would expect a quasi-linear line for each k until saturation, and this is indeed true for k<=3. However, for k=4, we see an increase until 4 layers, then a flat curve. Please justify why that is the case.

---

> ### Author Response · Authors · 2025-11-21
> **Rebuttal to the Reviewer TX7H**
>
> We sincerely thank the reviewer for the thoughtful feedback and positive comments regarding our experimental design, clarity, and the comprehensiveness of our study. We address each point below:
> 1. **Novelty and contribution beyond empirical analysis.** While we agree that the relationship between model depth and reasoning ability has been studied before, our contribution lies in presenting the comprehensive comparison of different architectures and depth extension mechanisms in terms of their reasoning abilities. Using our proposed 1dCA framework we show that recurrent memory (ARMT) and ACT are capable of adding ~100% (+1 step) computational depth in terms of cellular automata inference ot the small models, while reasoning training and teacher-forced autoregressive generation can add 200% (+2 steps) and at least 300% (+3 steps) respectively to the same models.
> 2. **Clarification on Chain-of-Thought (CoT) setup.**
> We appreciate the request for clarification. In our setup, CoT is implemented via training the model to predict intermediate reasoning traces, where each reasoning step is treated as an explicit CA state. This naming might be confusing as we don’t use the eponymous prompting technique. We will make this procedure clearer in the revised version.
> 3. **Explanation of the non-linear behavior for k=4 in Figure 4A.**
> We acknowledge the reviewer’s observation. The apparent plateau for k = 4 arises because the 1dCA rules at that step introduce longer-range dependencies that apparently exceed the model’s inference capabilities. While shallower models benefit from depth increases up to this limit, additional layers beyond 4 do not yield gains for such a deep task without explicit supervision on the reasoning steps. The particular difficulty of k=4 case lies in the fact that the space of possible state mappings from t to t+k grows exponentially with k, as the effective radius of dependencies (aka receptive field) grows linearly with k, making the task is significantly more difficult for larger k values. Moreover, if the model learns to apply the rule several times the state representation in hidden states may vanish making it harder to infer the rule even with sufficient model depth.
>
> We hope these clarifications make our contribution clearer.

---

> > ### Comment · Reviewer_TX7H · 2025-11-25
> > **Thank you for your rebuttal**
> >
> > I acknowledge the recipient of the rebuttal. The authors have adequately addressed my concerns in points 2 and 3. I still reserve my opinion that the work's novelty is mainly 1dCA and empirical results. Given that I'm not an expert in the field (indicated by my confidence), I'll choose to maintain the score and leave the judgment to other reviewers and AC.

---

### Official Review · Reviewer_qDUP · 2025-10-27

**Soundness:** 2
**Presentation:** 3
**Contribution:** 2
**Rating:** 4
**Confidence:** 4

**Summary:**

This paper uses a dataset constructed from one-dimensional cellular automata with controllable complexity to study the effects of model depth on reasoning ability. The authors discuss various architectures, including those incorporating recurrence and memory, and explore their reasoning capabilities within this stylized setting.

**Strengths:**

1. Clear writing with a good explanation of the setup.
2. A dataset with clearly controllable difficulty is nice for rigorous study.
3. It covers a diverse set of network architectures, which makes the study more convencing

**Weaknesses:**

There are two major weaknesses:
1. Lack of novelty. My main concern is the lack of novelty, as many similar stylized settings have been explored previously. I do not see any substantial new insights or conclusions beyond showing that the same qualitative story holds for additional architectures.
- Prior work [1] has already examined reasoning capability as a function of model depth while maintaining a comparable FLOPs budget, using a dataset with well-controlled difficulty and stronger resemblance to natural language.
- Work [2] also employed cellular automata datasets to study reasoning ability, showing a clear correlation between model capability and the complexity of the automata.
- Work [3] explicitly analyzed model depth in relation to data complexity under Wolfram’s classification, including skip step predictions.

2. Uncontrolled computational budget. Another major weakness lies in comparing different architectures without controlling for FLOPs or other measures of computational budget. This omission undermines the validity of the conclusions, since model width can also strongly affect performance. This issue was carefully discussed in [1].

[1] Physics of Language Models: Part 2.1, Grade-School Math and the Hidden Reasoning Process, https://arxiv.org/pdf/2407.20311

[2] Intelligence at the Edge of Chaos, https://arxiv.org/pdf/2410.02536v1

[3] Exploring Model Depth and Data Complexity Through the Lens of Cellular Automata, https://openreview.net/forum?id=SGoI97b5KK#discussion

**Questions:**

1. Could the authors address my concerns above?

2. From [2, 3, 4] it is clear that the complexity varies drastically in the cellular automata dataset, and predicting $t+n$ with $n>1$ is not always harder than $n=1$ due to the complex renormalization nature of the cellular automata. Could the authors discuss this further?


[4] Coarse-graining of cellular automata, emergence, and the predictability of complex systems, https://journals.aps.org/pre/abstract/10.1103/PhysRevE.73.026203

---

> ### Author Response · Authors · 2025-11-21
> **Rebuttal to the Reviewer qDUP**
>
> We sincerely thank the reviewer for their detailed comments and for recognizing our paper’s clear writing, dataset design, and comprehensive architecture coverage. Below we address each point in detail.
>
> **W1/Q1.** We appreciate the reviewer’s concern and agree that reasoning in stylized settings has been actively explored. However, our work introduces three concrete and novel contributions beyond:
> - 1dCA-Reasoning benchmark with separate rule induction and state propagation. Unlike [2] Intelligence at the Edge of Chaos and [3] Exploring Model Depth and Data Complexity, which analyze performance under shared rule sets or aggregated complexity classes, our 1dCA-Reasoning benchmark uses disjoint train/test rule sets, forcing genuine rule inference rather than memorization. This isolates abstraction and application – a distinction not made in prior CA studies.
> - LLM evaluation with natural-language proxy for the 1dCA benchmark. We introduce the Handsup dataset as a natural-language analogue to the 1dCA tasks and report the performance of several large language models. Despite their scale, these models fail to perform the underlying computation that can be solved by a compact 4-layer Transformer, highlighting the underutilisation of their computational capabilities. Such a memorizationless proxy was never presented before as far as we know.
> - Systematic depth-extension analysis under equal architectural conditions. Prior work [1] and [3] studied depth scaling within one architecture (transformer). In contrast, we standardize across four distinct model families (Transformers, RNNs, SSM, and RMTs) under identical hyperparameters and dataset complexity, providing the first controlled cross-architecture comparison of depth-extension mechanisms (recurrence, ACT, GRPO, CoT)
> - Depth extension interpretation in terms of depth of reasoning score. Our study explicitly analyzes test-time compute scaling (via ACT, recurrence and GRPO) as depth allocation strategies, demonstrating empirically that ACT provides ~+1 effective reasoning step in our toy setting, and RL extends this further to ~+2 steps . This connection between adaptive halting, reasoning horizon, and efficiency is unique to our paper. The aggregation of our results wrt depth of reasoning score can be found on Figure 6.
>
> **Q2.**
> > Complexity varies drastically in cellular automata… predicting t+n with n>1 is not always harder than n=1.
>
> We agree that prediction complexity depends on many factors such as initial state and rule complexity (see Figure 2(c)). However, in average predicting the further steps is harder, as we show it with the models’ performance degradation on Figure 5 (a,b,c). The challenge is not only in rule application, but also in understanding (during the training) that the rule should be inferred and applied several times. We believe our study is the first one analysing this challenge.
>
> **W2/Q1.**
> > comparing different architectures without controlling for FLOPs or other measures of computational budget.
>
>  The work did not focus on compute optimization, that’s why we didn’t restrict the models with the amount of computation they did. Instead, we observed the primary limitations of the architectures, which was the goal of this work.
> We appreciate the reviewer’s careful reading. Our revisions will emphasize the distinct contributions beyond existing CA-based reasoning studies and include an expanded discussion on CA complexity and computational fairness.
>
> We believe these clarifications demonstrate that the paper offers meaningful novelty, rigorous controls, and actionable insights into the mechanisms that extend reasoning depth.

---

### Official Review · Reviewer_3Utu · 2025-10-30

**Soundness:** 3
**Presentation:** 3
**Contribution:** 3
**Rating:** 4
**Confidence:** 3

**Summary:**

This paper investigates how large language models acquire and execute multi-step reasoning — specifically, whether their capability stems from memorization or true generalization. To study this, the authors design a new benchmark, 1dCA, which evaluates models’ ability to generalize beyond training distributions. Experimental results show that smaller models, such as standard Transformers, do exhibit non-trivial generalization, suggesting that their reasoning performance is not solely the result of memorized patterns.

**Strengths:**

1. The study tackles an important and timely question in reasoning research—whether LLMs’ multi-step reasoning ability reflects true generalization rather than mere memorization
2. Well-designed synthetic benchmarks: Allows controlled measurement and ablations not feasible on real datasets.
3. The analysis provides actionable guidance on how architectural design and training choices

**Weaknesses:**

In the Handsup experiments, simple CoT prompting is insufficient to reveal the model’s true reasoning capability. The experiments conducted on large models are not very convincing. For this type of task, the performance after SFT or in-context learning can differ substantially from zero-shot results. Moreover, Transformers achieve higher accuracy largely because they are trained on similar datasets, which does not demonstrate any fundamental difference between Transformers and LLMs, nor does it imply inherent limitations of LLMs. As a result, the experimental setup does not adequately support the subsequent interpretations regarding LLM behavior.

**Questions:**

It would be helpful to provide in-context learning results for LLMs on the Handsup benchmark. I suggest explicitly including the rule in the few-shot demonstrations to examine whether the model’s behavior changes when the underlying procedure is made more explicit.

---

> ### Author Response · Authors · 2025-11-21
> **Rebuttal to the Reviewer 3Utu**
>
> We thank the reviewer for the thoughtful feedback and for recognizing the importance of our study and the quality of our benchmark design and analysis. Below, we address each concern in detail and clarify points that may have led to misinterpretations.
>
> > Transformers achieve higher accuracy largely because they are trained on similar datasets, which does not demonstrate any fundamental difference between Transformers and LLMs.
>
> Our intention with the Handsup task was not to claim superiority of any model or show that small models are better reasoners than the current LLMs, but to motivate the controlled small-scale 1dCA benchmark. The Handsup results serve as a natural-language proxy demonstrating that even state-of-the-art LLMs fail on rule-based reasoning with disjoint rule sets (especially with radius=2). As you correctly noted, we show that small models trained on this specific reasoning dataset solve the task of one-step prediction perfectly, meaning that large models underutilise their computational abilities, and should probably be trained in a different way.
> Regarding the small-scale experiments, all models were trained on the same dataset from scratch, making the conditions equal for all architectures for fair comparison.
>
> > In the Handsup experiments, simple CoT prompting is insufficient to reveal the model’s true reasoning capability. The experiments conducted on large models are not very convincing. For this type of task, the performance after SFT or in-context learning can differ substantially from zero-shot results.
> …
> It would be helpful to provide in-context learning results for LLMs on the Handsup benchmark. I suggest explicitly including the rule in the few-shot demonstrations to examine whether the model’s behavior changes when the underlying procedure is made more explicit.
>
> Thank you for your suggestion to add the rule in the LLM prompt to assess the model’s ability to consistently apply it. This, however, is a simpler task, compared to the one we consider in this paper. Here our goal is specifically to check whether the model can both infer (or abstract) and apply the rule underlying the observations, mirroring our general understanding of reasoning tasks.
>
> Adding a few examples to the prompt can indeed increase the performance, but we don’t expect the difference to be dramatic. We show that models achieve below 10% accuracy with the rule radius 2, meaning that the problem is in the complete inability to perform complex rule inference. However, we appreciate your suggestion and will add this technique in the camera-ready version of the paper.
>
> We hope these points address the reviewer’s concerns and that the contribution – providing a clean benchmark, clear findings, and actionable insights on reasoning depth – will be recognized as both sound and valuable to the ICLR community.

---

### Note · Authors · 2026-01-06

**Comment:**

Applying for another conference.

**Withdrawal Confirmation:**

I have read and agree with the venue's withdrawal policy on behalf of myself and my co-authors.